# Adaptive Symmetry Discovery for Dynamical System Identification

**Behrooz Tahmasebi & Melanie Weber**
Harvard John A. Paulson School of Engineering and Applied Sciences (SEAS)
Harvard University, Cambridge, MA 02138, USA
`{behrooz_tahmasebi,mweber}@seas.harvard.edu`

## Abstract

Dynamical systems model trajectory data generated by underlying fixed dynamics, with applications ranging from biological systems to physics. Especially in scientific settings, dynamical systems are not generic, but exhibit symmetries imposed by physical laws, formalized as equivariance with respect to a group action. The identification problem concerns recovering the parameters of a system from observed trajectories. In this work, we study *adaptive symmetry discovery* for dynamical system identification and address how a system can be identified from a single trajectory when it is equivariant with respect to an unknown symmetry group. To this end, we first show that for known symmetries, the system can be identified from a significantly shorter single trajectory than in the generic setting, and we precisely characterize this improvement. We then consider the automatic symmetry discovery setting, proposing a method to learn the symmetry group directly from a single trajectory and incorporate it into the identification procedure, achieving the same optimal trajectory length as in the known-symmetry case. Our analysis relies on tools from group representation theory and the expander properties of Cayley graphs, and may be of independent interest for the study of symmetries in dynamical systems.

## 1 Introduction

The classical scientific approach has long relied on discovering fundamental laws of nature expressed through simple and interpretable equations. The recent abundance of data, together with rapid advances in machine learning and deep learning, has given rise to a complementary scientific paradigm: inferring governing laws and equations directly from data using principled methodologies rather than ad hoc modeling assumptions.

Among many modeling frameworks, dynamical systems constitute one of the most fundamental mathematical formalisms for describing flows, trajectory data, and non-independent evolutionary processes in the sciences, with applications ranging from biological systems to physical flows (Brunton et al., 2016; Yu & Wang, 2024; Wang & Yu, 2025). The problem of discovering governing equations from data, often formulated through the lens of dynamical systems (Brunton et al., 2016), has the potential to uncover previously unknown scientific principles and to significantly accelerate scientific discovery.

To this end, a central task is to learn the parameters of a dynamical system from data (observed trajectories), a problem commonly referred to as *dynamical system identification*. As a well-studied topic in system identification, a wide range of approaches and algorithms have been developed to address this problem across various settings (Van Overschee & De Moor, 2012; Isermann & Münchhof, 2011). In the context of scientific discovery from data, however, dynamical systems are often not generic. Instead, they are structured and frequently exhibit (potentially unknown) symmetries imposed by physical laws, formalized as equivariance with respect to a (possibly unknown) group action. The field of geometric machine learning studies how and when such symmetries can be exploited to improve learning and generalization (Bronstein et al., 2021; Weber, 2025). In this work, we focus on the intersection of geometric machine learning and dynamical system identification.

Motivated by these considerations, we study the problem of *adaptive symmetry discovery* for dynamical system identification. Focusing on single-trajectory data and finite groups, the central question we address is how to identify a system when it is equivariant with respect to an unknown symmetry group, and how to automatically discover and exploit this symmetry to improve sample efficiency, namely by enabling identification from shorter trajectories.

First, we show that when the symmetry group is known, the system can be identified from significantly shorter trajectories than in the generic setting, and we precisely characterize this improvement. Second, we turn to the automatic symmetry discovery setting and propose a method to learn the symmetry group directly from a single trajectory and to incorporate it into the identification procedure. Our approach achieves the same optimal trajectory length as in the known-symmetry case. Consequently, under mild conditions, symmetry discovery incurs negligible overhead, allowing one to fully leverage the benefits of equivariance.

It is worth noting that, while a number of recent studies are closely related to our work (as outlined in the related work section), most existing methods are heuristic or model-specific, and provable quantitative guarantees for symmetry discovery in dynamical systems are lacking. In contrast, our focus is on understanding the theoretical and foundational limits of this problem, which, to the best of our knowledge, have been largely unexplored in the literature.

We emphasize that the tools used in this paper, from group representation theory and the expansion properties of Cayley graphs for finite groups introduce techniques that are new to this context and may be of independent interest for the study of symmetries in dynamical systems and beyond.

## 2 PROBLEM STATEMENT AND MAIN RESULTS

**Dynamical systems.** A (discrete-time) dynamical system is specified by a function $f : \mathbb{R}^d \to \mathbb{R}^d$, which governs the state evolution according to $x_{t+1} = f(x_t), t = 0, 1, \ldots$, where $x_t \in \mathbb{R}^d$ denotes the *state* of the system at time $t$. We refer to $f$ as the *dynamics* of the system and assume that $f$ belongs to a prescribed function class $\mathcal{F}$.

**Trajectories and learning objective.** Given an initial state $x_0 \in \mathbb{R}^d$, the dynamics $f$ generates a *trajectory* $(x_0, x_1, \ldots, x_T) \in \mathbb{R}^{d \times (T+1)}$, where $T \in \mathbb{N}$ denotes the trajectory length. In the system identification problem, a learner observes such a trajectory and aims to recover the underlying dynamics $f \in \mathcal{F}$. Clearly, if the function class $\mathcal{F}$ is too rich, recovering $f$ from a finite-length trajectory is impossible. This motivates restricting attention to structured and low-complexity families of dynamics.

**Polynomially lifted linear dynamical systems** In this work, we focus on a class of dynamics that are linear after a polynomial lifting of the state.

**Definition 2.1** (Polynomially lifted linear dynamical systems)**.** *Let $\Phi : \mathbb{R}^d \to \mathbb{R}^m$ denote the feature map consisting of all monomials in $d$ variables $x = (x^1, x^2, \ldots, x^d)^\top \in \mathbb{R}^d$ of total degree at most $k \in \mathbb{N}$, that is, $\Phi(x) = \left( (x^1)^{\alpha_1}(x^2)^{\alpha_2} \cdots (x^d)^{\alpha_d} \right)_{\alpha \in \mathcal{I}_k} \in \mathbb{R}^m$, where $\mathcal{I}_k = \big\{ \alpha \in \mathbb{Z}_{\geq 0}^d \mid \sum_{i=1}^d \alpha_i \leq k \big\}$. The feature dimension is $m = \binom{d+k}{d}$. We now define $\mathcal{F}_k$, the set of polynomially lifted linear dynamical systems of degree at most $k$, as the class of functions of the form $f(x) = W\Phi(x)$, where $W \in \mathbb{R}^{d \times m}$.*

Thus, a polynomially lifted linear dynamical system is fully parameterized by a matrix $W \in \mathbb{R}^{d \times m}$. We refer to $\Phi$ as the *feature map*, mapping the state space to a higher-dimensional *feature space*. For convenience, we denote $\phi_t := \Phi(x_t) \in \mathbb{R}^m$. This modeling assumption encompasses a rich and expressive family of nonlinear dynamical systems while retaining a linear structure in feature space.

**Equivariant dynamics.** We now introduce the notion of symmetry in dynamical systems. Intuitively, a dynamical system is symmetric if transformations applied to its input state induce predictable and consistent transformations of its output state. For example, rotating the input state results in a correspondingly rotated output. Such structure reflects underlying invariances often imposed by physical laws and is pervasive in scientific and engineering applications.

**Groups and representations.**    A finite group $G$ is a finite set equipped with an associative binary operation, an identity element, and inverses for all elements. Common examples include finite rotation groups, reflection groups such as $\{\pm 1\}^d$, and the permutation group on $n$ elements, consisting of all bijections $\sigma : [n] \to [n]$ under composition.

A (linear) representation of a group $G$ on $\mathbb{R}^n$ is a homomorphism $\rho : G \to \mathrm{GL}_n(\mathbb{R})$, assigning to each $g \in G$ an invertible matrix $\rho(g) \in \mathbb{R}^{n \times n}$ such that $\rho(gh) = \rho(g)\rho(h)$, for all $g, h \in G$. We refer to $\rho(g)$ as the action of $g$ on $\mathbb{R}^n$.

**Lifted group actions.**    Suppose a finite group $G$ acts linearly on the state space $\mathbb{R}^d$ via a representation $\rho(g) \in \mathbb{R}^{d \times d}$. This action induces a natural action on the polynomial feature space associated with the lifting map $\Phi : \mathbb{R}^d \to \mathbb{R}^m$. Specifically, there exists a unique representation $\rho_\Phi : G \to \mathrm{GL}_m(\mathbb{R})$ such that $\Phi(\rho(g)x) = \rho_\Phi(g)\,\Phi(x)$, $\quad \forall g \in G, \ x \in \mathbb{R}^d$. The matrices $\rho_\Phi(g), g \in G$, form a representation of $G$ on the feature space $\mathbb{R}^m$.

**Definition 2.2** (Equivariant dynamical systems). *Let $G$ be a finite group acting on $\mathbb{R}^d$ via a representation $\rho$. A dynamical system $f : \mathbb{R}^d \to \mathbb{R}^d$ is said to be $G$-equivariant if and only if $f(\rho(g)x) = \rho(g)f(x)$, for all $g \in G$, $x \in \mathbb{R}^d$. For polynomially lifted linear dynamical systems $f(x) = W\Phi(x) \in \mathcal{F}_k$, $G$-equivariance is defined as $\rho(g)W = W\rho_\Phi(g)$, for all $g \in G$. We denote by $\mathcal{F}_k^G$ the class of $G$-equivariant polynomially lifted linear dynamical systems of degree at most $k$.*

**Definition 2.3** (Generic identifiability from a single trajectory). *Fix $k \in \mathbb{N}$ and a finite group $G$ and consider the class $\mathcal{F}_k^G$. The dynamics within this class are said to be (generically) identifiable from trajectories of length $T \in \mathbb{N}$ if, for almost all initial states $x_0 \in \mathbb{R}^d$ and almost all $G$-equivariant parameter matrices $W \in \mathbb{R}^{d \times m}$, the corresponding trajectory $(x_0, x_1, \dots, x_T) \in \mathbb{R}^{d \times (T+1)}$ uniquely determines $W$. The minimal such $T$ is denoted by $T(G) \in \mathbb{N}$.*

When identifiability holds for trajectories of length $T$, we informally refer to $T$ as the *sample complexity* of the identification problem, since each transition provides one observation of the dynamics. While our analysis focuses on polynomial feature maps, many of the ideas developed in this paper extend to other structured feature spaces, such as Fourier features. We adopt polynomial liftings as they provide a canonical and expressive class of nonlinear dynamics amenable to precise analysis.

**Symmetry discovery.**    Consider an unknown $G$-equivariant dynamical system where the underlying symmetry group $G$ is unknown. We assume, however, that $G$ belongs to a known class of admissible finite groups $\mathcal{G}$. While the specific group $G \in \mathcal{G}$ is not known a priori, prior knowledge of the class $\mathcal{G}$ provides structural information that can be leveraged for system identification. We are primarily interested in settings where $\mathcal{G}$ consists of relatively large finite groups, as such symmetries can lead to substantial reductions in the trajectory length required for identifiability. At the same time, the unknown identity of $G$ introduces a nontrivial challenge, as the learner must simultaneously identify the dynamics and discover the underlying symmetry.

The goal of *adaptive symmetry discovery* for dynamical system identification is to recover the dynamics $f$ from short trajectories, using only the assumption that $f$ is equivariant with respect to some unknown group $G \in \mathcal{G}$. In the ideal case, the best sample complexity one could hope to achieve is $T(\mathcal{G}) \coloneqq \max_{G \in \mathcal{G}} T(G)$, corresponding to the worst-case identifiability threshold over the class $\mathcal{G}$. A further challenge arises from computational considerations. Finite groups of interest are often prohibitively large. For example, the permutation group on $n$ elements has cardinality $n! \approx \exp(n \log n)$, as do many of its subgroups. Consequently, algorithms whose runtime scales linearly with the group size are infeasible, and one must instead aim for procedures with runtime polylogarithmic in the group size. Accordingly, our aim is to address the following question:

> *Given a class of groups $\mathcal{G}$ and an unknown dynamical system $f : \mathbb{R}^d \to \mathbb{R}^d$ that is $G$-equivariant for some unknown $G \in \mathcal{G}$, can one identify the dynamics using trajectories of length $T(\mathcal{G})$ while maintaining an efficient computational runtime?*

**Irreducible representations.**    Let $G$ be a finite group, and let $\rho : G \to \mathbb{C}^{n \times n}$ be a linear representation. The representation $\rho$ is called *irreducible* if there exists no nontrivial invariant subspace, or

equivalently, if the matrices $\{\rho(g)\}_{g \in G}$ cannot be simultaneously block-diagonalized by any change of basis. The set of (equivalence classes of) irreducible representations of $G$ is finite and is denoted by $\widehat{G}$. Any finite-dimensional representation $\rho$ admits a decomposition, after an appropriate change of basis, into a direct sum of irreducible representations: $\rho \cong \bigoplus_{\pi \in \widehat{G}} n_\pi \pi$, with multiplicities $n_\pi \in \mathbb{Z}_{\geq 0}$.

**Theorem 2.4** (Equivariant system identification). *Fix a finite group $G$, and consider polynomially lifted linear dynamical systems $f(x) = W\Phi(x) \in \mathcal{F}_k^G$. Let $\rho_\Phi \cong \bigoplus_{\pi \in \widehat{G}} n_\pi \pi$ denote the irreducible decomposition of the lifted representation on the feature space, where $n_\pi$ is the multiplicity and $d_\pi$ is the dimension of $\pi \in \widehat{G}$. Then the sample complexity of $G$-equivariant system identification is given by $T(G) := \max_{\pi \in \widehat{G}} \left\lceil \frac{n_\pi}{d_\pi} \right\rceil \in \mathbb{N}$.*

Theorem 2.4 admits a simple and instructive interpretation. Since $d_\pi \geq 1$ and $n_\pi \leq m$ for all $\pi \in \widehat{G}$ (where $m$ is the feature dimension), we always have the trivial upper bound $T(G) \leq m$, for all $G$. This bound is tight in the absence of symmetry, i.e., when $G$ is the trivial group. However, the presence of symmetry can dramatically reduce the required trajectory length. Indeed, representation theory implies the identity $\sum_{\pi \in \widehat{G}} n_\pi d_\pi = m$, which already suggests that the ratios $n_\pi/d_\pi$ can be much smaller than $m$ when high-dimensional irreducible components are present.

**Adaptive symmetry discovery.** We now turn to the problem of identifying equivariant dynamical systems when the underlying symmetry group is unknown. We recall the notion of generating sets, which allows for a compact representation of finite groups.

**Definition 2.5** (Generating set). *A subset $S \subseteq G$ of a finite group $G$ is called a generating set if every element $g \in G$ can be expressed as a finite product of elements of $S$, that is, $g = s_1 s_2 \cdots s_n$ for some $s_i \in S$. In this case, we write $G = \langle S \rangle$.*

Generating sets provide a succinct description of potentially large groups and play a central role in our symmetry discovery procedure. Unfortunately, finding a smallest generating set of a finite group is computationally difficult; this motivates the use of expander constructions to obtain reasonably small generating sets.

**Theorem 2.6** (Adaptive symmetry discovery). *Consider the problem of identifying an unknown dynamical system $f$ that is equivariant with respect to an unknown finite group $G \in \mathcal{G}$, and $f \in \mathcal{F}_k^G$, for some $k \in \mathbb{N}$. Let $|G|_{\max} := \max_{G \in \mathcal{G}} |G|$, and fix a failure probability $\delta \in (0, 1)$. There exists an algorithm (Algorithm 1) such that, given access to a single trajectory $(x_0, x_1, \ldots, x_T) \in \mathbb{R}^{d \times (T+1)}$ generated from a generic initial state by a generic $G$-equivariant system, the algorithm recovers the system and the symmetry group with probability at least $1 - \delta$ (over the algorithm's randomness), provided that $T \geq T(\mathcal{G}) := \max_{G \in \mathcal{G}} T(G)$. Moreover, the algorithm returns a generating set $S$ such that $G = \langle S \rangle$. The runtime is polynomial in $m$, $|\mathcal{G}|$, $\log \frac{1}{\delta}$, and $\log |G|_{\max}$.*

Algorithm 1 is probabilistic and depends on a failure probability $\delta \in (0, 1)$. It requires the ability to sample uniformly at random from the candidate groups $G \in \mathcal{G}$. The failure probability can be made arbitrarily small by increasing the number of sampled group elements, which scales logarithmically in $1/\delta$. Moreover, the algorithm can be made deterministic, with no probability of failure, if a small (i.e., logarithmic-size) generating set is given for each $G \in \mathcal{G}$. This case is simpler to handle as it follows directly from the arguments presented here (proof omitted).

The algorithm achieves the *optimal sample complexity* for equivariant identification: as long as $T \geq T(\mathcal{G})$, a generic $G$-equivariant system can be identified. In addition, the algorithm successfully discovers the unknown symmetry by returning a generating set $S$ such that $G = \langle S \rangle$. Finding such an $S$ is essential for understanding how to project onto the space of $G$-equivariant dynamical systems, and allows one to fully test whether the dynamics is $G$-equivariant.

Executing Algorithm 1 involves checking feasibility of linearly constrained systems of linear equations and computing a solution whenever one exists. As shown in the proof, these tasks can be carried out in time polynomial in $(m, T, \log |G|_{\max})$. Importantly, the dependence on the group size is only logarithmic, which is essential for scalability.

Equivariant dynamical systems can be identified from significantly shorter trajectories than those required in the generic, non-equivariant setting. This reduction can be fully realized even when the symmetry group is unknown: the underlying symmetry can be discovered adaptively with only polylogarithmic computational overhead in the group size.

## ACKNOWLEDGMENTS

BT and MW were supported by NSF awards CBET-2112085 and DMS-2406905. MW also acknowledges support from an Alfred P. Sloan Fellowship in Mathematics and an Aramont Fellowship for Emerging Science Research.

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

## A  RELATED WORK

Recently, the problem of learning dynamical systems from data has received particular attention, driven by applications across the sciences and physics-guided machine learning (Brunton et al., 2016; Yu & Wang, 2024; Wang & Yu, 2025; Sivaranjani et al., 2025). From a theoretical perspective, characterizing when and how a dynamical system can be identified from data is a classical and well-studied problem in system identification (Van Overschee & De Moor, 2012; Isermann & Münchhof, 2011). More recently, significant effort has focused on learning dynamical systems from noisy observations, a setting that goes beyond, and is substantially more challenging than, the noiseless identification regime. For example, Simchowitz et al. (2018) study the identification of linear dynamical systems from a single noisy trajectory, with subsequent extensions to nonlinear systems under similar single-trajectory assumptions (Foster et al., 2020; Ziemann et al., 2022).

Beyond this, prior work has also investigated finite-sample identification of linear time-invariant systems (Sarkar & Rakhlin, 2019; Sarkar et al., 2021), regimes involving multiple trajectories (Tu et al., 2024), as well as system identification and control over a single trajectory (Fefferman et al., 2022; Carruth et al., 2022; 2024). In contrast, the focus of this paper is on the identification of dynamical systems *simultaneously* with symmetry discovery. As a first step toward this goal, we focus on the noiseless single-trajectory setting, which allows us to study the fundamental role of symmetry without additional statistical complications.

Although known symmetries are known to provide both empirical and theoretical benefits in learning and estimation, including improved generalization guarantees (Tahmasebi & Jegelka, 2023; 2024), in many applications the relevant symmetries exist but are not known *a priori*. In such settings, symmetries must be detected or discovered directly from data, as commonly encountered in the discovery of physical laws governed by differential equations. The research direction of *automatic symmetry discovery* seeks to identify underlying symmetries in a principled manner, moving beyond ad hoc or manually engineered approaches.

A wide range of methods for symmetry discovery have been proposed in the literature. Deep learning approaches for symmetry discovery have been explored in several works (Desai et al., 2022; Yang et al., 2023; Perin & Deny, 2025), including Lie algebra–based convolutional networks (Dehmamy et al., 2021); see also (Romero & Lohit, 2022; Ko et al., 2024; Hu et al., 2025b). Methods based on infinitesimal generators for discovering symmetries in nonlinear dynamical data have also been studied (Hu et al., 2025c); see (Shaw et al., 2025) for related approaches.

Beyond these, methods for discovering nonlinear group actions in latent spaces have been proposed (Yang et al., 2024a), including approaches that go beyond affine transformations and address manifold-structured data (Shaw et al., 2024; Bhat et al., 2025). Another line of work learns symmetries from layer gradients to relax hard invariance constraints (van der Ouderaa et al., 2023), while flow matching has recently been used to discover Lie group symmetries (Park et al., 2025). For symmetry discovery in differential equations and PDEs, see (Kreider et al., 2025; Hu et al., 2025a; Yang et al., 2025; 2024b). Other approaches include quadratic-form-based methods (Karjol et al., 2025) and learnable data augmentation strategies (Santos-Escriche & Jegelka, 2025). For symmetry discovery in finite groups using representation-theoretic tools, see (Huh, 2025). Jointly discovering and enforcing symmetries has also been studied (Otto et al., 2025). Finally, we note that symmetry discovery is fundamentally different from (binary) hypothesis testing for the presence of symmetry in data (Soleymani et al., 2025b).

There has also been recent work on symmetry discovery, specifically in dynamical systems. Data-driven detection of Lie point symmetries for continuous dynamical systems is studied in Gabel et al. (2024), while the discovery of finite symmetry groups in dynamical systems is considered in Calvo-Barlés et al. (2025a;b). A related approach is proposed in Li et al. (2025), where the authors introduce latent mixtures of symmetries to model dynamical systems with multiple symmetric latent components.

Finally, we note that a variety of approaches have been proposed to introduce and exploit symmetries in learning, ranging from canonicalization (Kaba et al., 2023; Tahmasebi & Jegelka, 2025a;b; Shumaylov et al., 2025) and frame averaging (Puny et al., 2022; Atzmon et al., 2022; Lin et al., 2024) to data augmentation (Tahmasebi et al., 2025). In contrast, this paper introduces an approach based on generating sets of groups and their expansion properties, with close connections to recent work

on approximate symmetry (Tahmasebi & Weber, 2025) and learning under invariances (Soleymani et al., 2025c;a).

## B   Proof Sketch

In this section, we provide a proof sketch of Theorem 2.6. We begin by examining the structure of Algorithm 1. The algorithm treats the observed trajectory as $T$ evaluations of the unknown dynamical system $f : \mathbb{R}^d \to \mathbb{R}^d$, which satisfies

$$x_{t+1} = W\Phi(x_t), \qquad t = 0, 1, \dots, T-1. \tag{1}$$

Letting $\phi_t = \Phi(x_t)$ for all $t$, we define the data matrices

$$X = [\phi_0, \phi_1, \dots, \phi_{T-1}] \in \mathbb{R}^{m \times T},$$

$$Y = [x_1, x_2, \dots, x_T] \in \mathbb{R}^{d \times T}.$$

With this notation, identifying the dynamics reduces to finding a matrix $W \in \mathbb{R}^{d \times m}$ satisfying the linear system

$$WX = Y. \tag{2}$$

Any solution to Equation 2 corresponds to a dynamical system that is perfectly consistent with the observed trajectory. Consequently, the central questions are whether such a solution exists and, if so, whether it is unique.

Existence is immediate: by assumption, the trajectory is generated by a polynomially lifted linear dynamical system, and hence at least one solution $W$ must satisfy Equation 2. Uniqueness, however, is more subtle. In the generic (non-equivariant) setting, one might hope that $X$ is invertible, allowing the recovery $W = YX^{-1}$. This would require $T \geq m$, which is precisely the regime we seek to avoid. In the presence of symmetry, we aim to identify the system from trajectories of much shorter length.

To characterize all solutions of Equation 2, we invoke standard linear algebra. Let $X^\dagger \in \mathbb{R}^{T \times m}$ denote the Moore–Penrose pseudoinverse of $X$, which always exists and is unique. Then the set of all solutions to Equation 2 is given by

$$W = YX^\dagger + Z(I_m - XX^\dagger), \qquad Z \in \mathbb{R}^{d \times m}, \tag{3}$$

where $Z$ is an arbitrary matrix. The term $YX^\dagger$ is the minimum-norm solution, while the second term spans the null space of the linear map $W \mapsto WX$. As $Z$ varies, Equation 3 parameterizes all matrices $W \in \mathbb{R}^{d \times m}$ consistent with the observed data.

The key insight underlying Algorithm 1 is that equivariance constraints significantly restrict the admissible set of matrices $W$ in Equation 3. By intersecting the affine solution space of Equation 2 with the linear subspace imposed by equivariance, the degrees of freedom collapse, yielding identifiability from short trajectories. The remainder of the proof formalizes this intuition and shows that sampling group elements suffices to recover the unknown symmetry with high probability.

For equivariant systems, it is generally the case that $XX^\dagger \neq I_m$, and therefore the solution to the linear system $WX = Y$ is not unique. In this regime, equivariance plays a crucial role. Since the true dynamics is equivariant with respect to a group $G$, the unknown matrix $W$ must satisfy

$$\rho(g)W = W\rho_\Phi(g), \qquad \forall g \in G. \tag{4}$$

Each such condition is linear in the entries of $W$, and collectively they impose additional constraints that can eliminate the spurious degrees of freedom in the solution set of $WX = Y$.

A direct implementation of these constraints, however, is computationally infeasible: enforcing equivariance for all $g \in G$ would require solving a system with $|G|$ linear constraints, which is prohibitive when $G$ is large. To overcome this issue, we leverage the notion of generating sets.

If $S \subseteq G$ is a generating set of $G$, written as $G = \langle S \rangle$, then equivariance with respect to $S$ already implies equivariance with respect to the entire group:

$$\rho(g)W = W\rho_\Phi(g), \quad \forall g \in S$$
$$\implies \rho(g)W = W\rho_\Phi(g), \quad \forall g \in G.$$

---

**Algorithm 1** Adaptive symmetry discovery

---

1: **Input:** trajectory $(x_0, x_1, \ldots, x_T) \in \mathbb{R}^{d \times (T+1)}$, degree $k \in \mathbb{N}$ (or feature dimension $m = \binom{d+k}{d}$), failure probability $\delta \in (0, 1)$

2: **Output:** $W \in \mathbb{R}^{d \times m}$ and a generating set $S$ such that the dynamics is equivariant to $G = \langle S \rangle$

3: Compute features $\phi_t \leftarrow \Phi(x_t)$ for all $t = 0, \ldots, T-1$, and form

$$X \leftarrow [\phi_0, \phi_1, \ldots, \phi_{T-1}] \in \mathbb{R}^{m \times T},$$

$$Y \leftarrow [x_1, x_2, \ldots, x_T] \in \mathbb{R}^{d \times T}.$$

4: **for** each $G \in \mathcal{G}$ **do**

5:     Set

$$N \leftarrow 2.67 \left( \log |\mathcal{G}| + \log |G|_{\max} + \log \tfrac{1}{\delta} + \log 2 \right),$$

$$S \leftarrow \{g_1, \ldots, g_N\}, \quad g_i \overset{\text{iid}}{\sim} \text{Unif}(G).$$

6:     **Feasibility test:** check whether there exists $W \in \mathbb{R}^{d \times m}$ such that

$$WX = Y \quad \text{s.t.} \quad \rho(g)W = W\rho_\Phi(g), \ \forall g \in S.$$

7:     **if feasible then**

8:         Return $S$ and any feasible solution $W$, and terminate.

9:     **end if**

10: **end for**

---

This implication follows directly from the definition of a generating set. Consequently, once a generating set $S$ is identified, imposing equivariance constraints only for elements of $S$ suffices to recover the unique $G$-equivariant solution $W$.

A key observation is that feasibility of the constrained system provides a test for equivariant to the unknown group $G$. Specifically, for any group $G \in \mathcal{G}$ with a generating set $S \subseteq G$, the system

$$WX = Y \quad \text{s.t.} \quad \rho(g)W = W\rho_\Phi(g), \quad \forall g \in S$$

is feasible if and only if the system is $G$-equivariant, provided that $T \geq T(\mathcal{G})$. This allows us to determine, using data alone, whether a sampled element belongs to the true symmetry group.

The core idea behind adaptive symmetry discovery is therefore as follows. We sample a collection of elements from any group $G \in \mathcal{G}$ to form a subset $S \subseteq G$ as a proxy for a generating set and then test each group for feasibility. When the number of samples $N$ is sufficiently large, we encounter elements of the subset $S \subseteq G$ form a generating set, with high probability.

Indeed, classical results on Cayley graph expanders (Alon & Roichman, 1994) imply that a random subset of a finite group $G$ of size $\mathcal{O}(\log |G| + \log \tfrac{1}{\delta})$ forms a generating set with probability at least $1 - \delta$. Combining this fact with the feasibility test described above and using a union bound shows that sampling

$$N = \mathcal{O}\left( \log |\mathcal{G}| + \log |G|_{\max} + \log \tfrac{1}{\delta} \right)$$

is sufficient to recover a generating set of the unknown symmetry group $G$, which in turn enables identification of the dynamics from short trajectories.

## C   EXAMPLE: QUADRATIC SYSTEMS WITH PERMUTATION SYMMETRY

Consider a quadratic dynamical system ($k = 2$) on $\mathbb{R}^d$ with permutation equivariance under the symmetric group $S_d$, where $d \geq 4$. The feature space of polynomials of degree at most two decomposes as

$$\mathcal{P}_{\leq 2} = V_0 \oplus V_1 \oplus V_2,$$

where $V_i$ denotes the space of homogeneous polynomials of degree $i$. The irreducible decompositions of these spaces are given by

$$V_0 = \pi_0,$$
$$V_1 = \pi_0 \oplus \pi_{\text{std}},$$
$$V_2 = 2\pi_0 \oplus 2\pi_{\text{std}} \oplus \pi_{(d-2,2)},$$

where $\pi_0$ is the trivial representation, $\pi_{\text{std}}$ is the standard representation with $d_{\pi_{\text{std}}} = d - 1$, and $\pi_{(d-2,2)}$ has dimension $d(d-3)/2$.

Consequently, the multiplicities are

$$n_{\pi_0} = 4, \qquad n_{\pi_{\text{std}}} = 3, \qquad n_{\pi_{(d-2,2)}} = 1,$$

and we obtain

$$T(G) = \max\left\{4, \left\lceil \frac{3}{d-1} \right\rceil, \left\lceil \frac{2}{d(d-3)} \right\rceil\right\} = 4.$$

In contrast, the feature dimension in this setting satisfies

$$m = \sum_{\pi \in \widehat{G}} n_\pi d_\pi = 4 + 3(d-1) + \frac{d(d-3)}{2} = \Theta(d^2).$$

Thus, while $\Theta(d^2)$ trajectory length is required in the absence of symmetry, permutation equivariance reduces the sample complexity to a constant, independent of the state space dimension $d$.

This example illustrates the substantial benefits of symmetry in dynamical system identification. Achieving the optimal bound $T(G)$ without prior knowledge of the symmetry group $G$, which is the focus of the paper, is both challenging and highly appealing.

