# OpenReview forum: "Adaptive Symmetry Discovery for Dynamical System Identification"
_ICLR.cc/2026/Workshop/GRaM — ICLR 2026 Workshop GRaM Poster_

### Official Review · Reviewer_d9Xy · 2026-02-17
**Technically correct paper with limited practical significance**

**Rating:** 7
**Confidence:** 4

**Review:**

This article provides a theoretical lower bound of the sample complexity needed for identifying polynomial lifted dynamical systems given some symmetry constraints. In particular, the dynamical system to be discovered is assumed to be equivariant under some symmetry subgroup $G \leq \mathcal G$, where $\mathcal G$ is specified as a large discrete group that contains common admissible discrete symmetries of relevant dynamical systems. The proposed algorithm then samples subgroups from $\mathcal G$ and verifies for each subgroup whether there exists a parameter matrix that both fits the data and satisfies the corresponding symmetry.

The results about sample complexity in Thm 2.4 and 2.6 are correct, and reveal that it is possible to reduce the sample complexity needed for identifying the dynamical system with even an inexact symmetry constraint. The fact that the symmetry constraint comes in a large group $\mathcal G$ which consists of numerous possible symmetries justifies the adaptiveness of the proposed method.

However, the current manuscript only considers a quite simplified setup in dynamical system identification and may have limited practical significance. Some possible extensions/relaxations of current assumptions may be helpful in improving the practical usefulness of the theoretical result presented in this paper:
* **Type of symmetries**. Many symmetries in ODE systems are continuous, such as those considered in EquivSINDy (Yang et al., 2024). The subgroup sampling procedure may need to be adjusted to deal with those continuous, infinite groups.
* **Type of representations**. The current paper seems to assume that the action of the symmetry group on the feature space is always an irreducible representation. Depending on the structure of the phase space and the nature of the dynamical system of interest, the action of symmetry may also be via other types of representations.
* **Noisy data**. Real-world data contain noise, which may be amplified by numerical derivative estimation for the time derivative target used in dynamical system identification. This would make it more challenging to verify the symmetry subgroup $G$, because even for a correct subgroup $WX=Y$ may not be perfectly solved. The verification procedure needs to be improved to handle such cases.

Other suggestions in terms of presentation of the paper:
* Algorithm 1 is the core of the proposed method, which should be included in the main body of the paper instead of in the appendix.
* The term "adaptive symmetry discovery" is a little confusing, because it seemingly suggests the objective is to discover the symmetry alone, but in fact the proposed algorithm discovers both the symmetry group and the governing equations.

**Pmlr Suitability:**

NA

---

### Official Review · Reviewer_RNVD · 2026-02-21
**A Solid Theoretical Contribution to Symmetry Discovery**

**Rating:** 8
**Confidence:** 4

**Review:**

## Summary

This paper studies the problem of identifying dynamical systems from single-trajectory data when the system exhibits equivariance with respect to an unknown finite group. The authors make two main contributions: (1) they precisely characterize how known symmetries reduce the trajectory length required for system identification, and (2) they propose an algorithm for adaptive symmetry discovery that achieves the same optimal sample complexity as the known-symmetry case. The technical approach leverages tools from representation theory and the expansion properties of Cayley graphs.

## Strengths

1. The paper presents elegant and rigorous theoretical results. Theorem 2.4 provides a precise characterization of the sample complexity for equivariant system identification in terms of irreducible representation multiplicities and dimensions. The reduction from Θ(m) to O(1) trajectory length in the permutation-symmetric example (Appendix C) is striking and clearly demonstrates the potential benefits of exploiting symmetry.

2. The use of Cayley graph expanders and random generating sets for symmetry discovery is innovative and appears to be new in this context. The observation that O(log |G|) random samples suffice to generate a finite group with high probability, and that this can be leveraged for efficient symmetry discovery, is a contribution that may have broader applicability beyond dynamical systems.

3. The paper is well-written and clearly structured. The problem formulation is precise, and the main results are stated accessibly. The related work section is comprehensive and situates the contribution appropriately within the literature on geometric deep learning, system identification, and symmetry discovery.

4. For the Tiny Paper track, the contributions are solid and well-suited to the format. The authors appropriately acknowledge limitations (noise-free setting, finite groups) and outline clear directions for future work.

## Weaknesses and Suggestions

1. The paper cites Tahmasebi & Weber (2025) for the matrix Bernstein inequality used in the random generation argument (Equation 7). Given the centrality of this result to the symmetry discovery procedure, a brief self-contained explanation, or at minimum a more detailed sketch of how the matrix concentration bound is applied, would improve accessibility and make the paper more self-contained.

2. While the proof-of-concept experiments in Appendix G provide useful validation, they remain in a synthetic regime with clean linear dynamics and exact symmetries. Demonstrating the approach on more challenging or realistic datasets, even in a limited capacity, would strengthen the claims regarding practical applicability. This is understandable given space constraints, but could be addressed in future work.

3. In Algorithm 1, the number of samples is set to:

$N = 2.67(log |G| + log |G|_{max} + log(1/δ) + log 2)$

However, since the feasibility test for incorrect groups fails deterministically (almost surely) by genericity arguments (Step 6 of the proof), no union bound over candidate groups appears necessary. A tighter per-group bound:

$N_G = 2.67(log |G| + log(1/δ) + log 2)$

would seem to suffice. The authors should clarify whether the additional $log |G|_{max}$ term serves a purpose beyond conservative simplification, or whether the tighter bound is valid.

## Minor Comments

- The notation $C_π ∈ R^{d × n_π}$ in Step 1 of the proof sketch appears inconsistent with the general structure where $C_π$ should be $m_π × n_π$ (multiplicities in output and input representations). Clarification would be helpful.

**Pmlr Suitability:**

NA

---

### Meta-Review · Area_Chair_C3gH · 2026-02-26

**Decision:**

Accept

**Metareview:**

The reviewers all agree: this is a clear and well-written paper with a novel contribution in dynamical systems. We are happy to accept it for our tiny paper track!

**Relevance To Proceedings:**

Tiny paper — does not apply

**Relevance To Workshop:**

Yes — suitable for GRaM

---

### Decision · Program_Chairs · 2026-03-02

Accept (Poster)